

# Global marine phytoplankton dynamics analysis with machine learning and reanalyzed remote sensing

Subhrangshu Adhikary[1], Surya Prakash Tiwari[2], Saikat Banerjee[3], Ashutosh Dhar Dwivedi[4] and Syed Masiur Rahman[2]

[1] Spiraldevs Automation Industries Pvt. Ltd., Raiganj, West Bengal, India
[2] King Fahd University of Petroleum and Minerals (KFUPM), Dhahran, Saudi Arabia
[3] Wingbiotics, Baghajatin, Kolkata, West Bengal, India
[4] Cybersecurity Section, Aalborg University, Copenhagen, Denmark

## ABSTRACT

Phytoplankton are the world's largest oxygen producers found in oceans, seas and large water bodies, which play crucial roles in the marine food chain. Unbalanced biogeochemical features like salinity, pH, minerals, *etc.*, can retard their growth. With advancements in better hardware, the usage of Artificial Intelligence techniques is rapidly increasing for creating an intelligent decision-making system. Therefore, we attempt to overcome this gap by using supervised regressions on reanalysis data targeting global phytoplankton levels in global waters. The presented experiment proposes the applications of different supervised machine learning regression techniques such as random forest, extra trees, bagging and histogram-based gradient boosting regressor on reanalysis data obtained from the Copernicus Global Ocean Biogeochemistry Hindcast dataset. Results obtained from the experiment have predicted the phytoplankton levels with a coefficient of determination score ($R^2$) of up to 0.96. After further validation with larger datasets, the model can be deployed in a production environment in an attempt to complement *in-situ* measurement efforts.

## INTRODUCTION

Phytoplankton are microscopic, single-celled photosynthetic organisms that live in both fresh and salty water environments. They have photosynthetic pigments which significantly contribute to global oxygen production. The chlorophyll pigments found in phytoplankton absorb the solar radiation and convert the solar energy into chemical energy through the photosynthesis process. Chlorophyll pigments are often used as an indicator of water quality as well as phytoplankton biomass (*Rizzuto et al., 2020*). Therefore, phytoplankton plays an important role in maintaining the marine ecological balance by becoming a major portion of the marine food chain. Phytoplankton need a suitable environment for optimum growth. The imbalance in the concentration of different biogeochemical features affects phytoplankton growth (*Adhikary et al., 2021*; *Chai et al., 2021*). These features are currently available and can be obtained from remote sensing sensors by processing the reflectance spectrum.

Corresponding author
Ashutosh Dhar Dwivedi,
addw@es.aau.dk

Several biological and chemical properties of the oceans like phytoplankton, primary productivity, dissolved nitrites, iron, oil, and several other parameters could be observed using remote sensing technologies (*Sun et al., 2022*; *Adhikary, Tiwari & Banerjee, 2022*). Many of these properties can be detected through the light of specific wavelengths. For example, 550–560 and 700–719 nm range can be studied to detect phytoplankton (*Kramer et al., 2022*). Moreover, different ratios of remote sensing reflectance (RRS) can be utilized to observe certain properties. For example, Chl-a can be best detected from R443/R490 and R490/R555 configurations (*Kolluru & Tiwari, 2022*). Further, *in-situ* observations can be used to fine-tune the results obtained from the experiments.

Active fluorescence methods have been extensively used around the world for estimation of the primary production normalized to Chl-a and used regression algorithms to analyze the yield slope (*Seenipandi et al., 2021*). Remote sensing technologies have been used to study the sun-induced chlorophyll fluorescence and understand the natural variations in the optical characteristics of phytoplankton discussed (*Hu et al., 2021*; *Liu et al., 2021a*; *Pahlevan et al., 2021*; *Xu et al., 2021*). Figure 1 shows the distribution of global phytoplankton concentration and the number marks of the study locations for the experiment which have been discussed later in this study. Ocean colour remote sensing has significantly advanced in the last four decades since the launch of the first ocean colour sensor the Coastal Zone Color Scanner Experiment (CZCS) (*McClain, Franz & Werdell, 2022*). Advanced ocean colour sensors are currently available that can be used for monitoring phytoplankton distribution remotely at a global scale. The phytoplankton blooms cause a change in ocean colour that can be observed from the space (*Zhong et al., 2021*). Iron distribution has a linear dependency on the phytoplankton biomass (*Gomaa et al., 2021*; *Thomalla et al., 2021*). There also exists a nonlinear relationship between dissolved cadmium and phosphate and the phytoplankton growth affecting the phytoplankton size (*Schine et al., 2021*; *Zilius et al., 2021*). Further phytoplankton growth is dependent upon the presence of salinity, nutrients, light, pH and turbulence (*Sun et al., 2021*; *Wang et al., 2021a*)

The biogeochemical features obtained from the remote sensing sensors can be processed with machine learning algorithms for smarter decision-making. Currently, there are several machine learning regression algorithms available. Random forest is a well-known supervised machine learning algorithm that learns by constructing several trees and deciding the outcome based on votes cast by the majority of trees. It is well-known for producing consistent results, and it has been widely used to detect algal blooms using remote sensing data (*Benmokhtar et al., 2021*). Bagging regressor is an ensemble algorithm that fits the base regressor to random subsets of the training dataset by substituting some subsets with others. It has earlier been used for the classification of corals for oceanographic surveys (*Gómez et al., 2021*). Extra trees regressor also known as extremely randomized trees regressor is a type of supervised learning algorithm which works by fitting on extremely randomized decision trees. It has earlier been used in several domains such as to retrieve optically active parameters from oceanic reflectance spectra (*Turkmen, Chee & Huff, 2021*). Finally, a Histogram-based gradient boosting regressor is a form of ensembled decision tree and works extremely fast for large datasets (*Guryanov, 2019*).

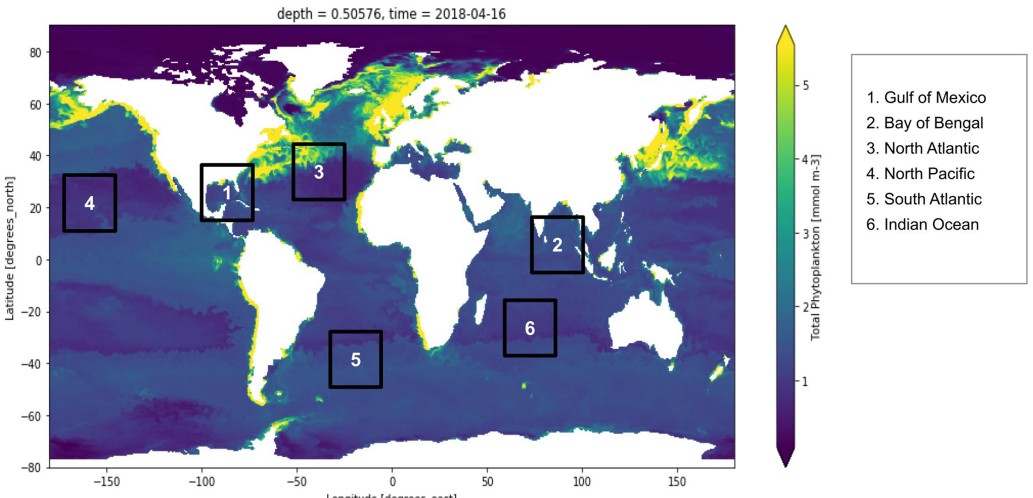

**Figure 1 Global marine phytoplankton distribution at a depth of 0.5 m, on 16th April 2018.**
The bluish region indicates low phytoplankton concentration and the yellow region indicates higher phytoplankton concentration. Six points have been marked which are the study locations considered for the experiment. These six locations are Gulf of Mexico, Bay of Bengal, North Atlantic, North Pacific, South Atlantic and Indian Ocean.

Reanalysis of remote sensing with calibration from *in-situ* observations to produce a robust dataset is a very promising and growing technique for large-scale ocean monitoring (*Ma et al., 2021*). NASA's ocean colour processing group is working on developing a new algorithm for satellite estimation. However, artificially intelligent algorithms such as machine learning (ML) and deep learning (DL) have not been extensively investigated for estimating phytoplankton concentration (*Wang et al., 2021b*; *Grøtte et al., 2021*). Because AI-based approaches have been widely implemented in a variety of other domains, these methods can also be used to monitor the colour of water bodies (*Puissant et al., 2021*; *Liu et al., 2021b*). These models will be very useful for remotely monitoring the health of the marine ecosystem and making smart decisions (*Asim et al., 2021*). This study aims to: i) develop and test supervised learning models on six different locations around the world using the Copernicus Global Ocean Biogeochemistry Hindcast and Physical GLOBAL REANALYSIS datasets, and ii) remotely predict phytoplankton concentration based on other biogeochemical features. We used four ML algorithms in this study: random forest regressor, bagging regressor, extra trees regressor, and histogram-based gradient boosting regressor (HGBR).

## Research background

As per recent studies conducted by *Groom et al. (2019)* ocean colour data plays a vital role in determining various types of oceanic phenomena such as harmful algal bloom, coastal eutrophication, sediment plumes to observe and analyse global scale. Also by using various specialised optical sensors like OLCI, MERIS, Aqua-MODIS, VIIRS, and Sea-WiFS the observation assessments can be done by decade statistics focused on medium to large resolution based ocean and coastal sea operations. The global climate observing system (GCOS) acknowledges the ocean colour as an essential climate variable (ECV) specifically

in the visible domain, along with chlorophyll-a, which is also identified as one of the required products for ECV (*Sathyendranath et al., 2019*). Recent scientific assessments show that the reflectance of the 443 nm band was present on most ocean colour sensors for the peak in chlorophyll-a specific absorption while the chlorophyll-a specific absorption at 412 nm reaches about 70% of that at 443 nm (*O'Reilly & Werdell, 2019*).

Several machine learning and deep learning algorithms have been used in the literature for analysing remote sensing data. Novel approaches with random forest-based regression improve spatial and temporal coverage of chlorophyll detection using a combination of high-quality, low-coverage, chlorophyll and lower-quality remote sensing data (*Chen et al., 2019*). This helps in global chlorophyll-a mapping with up to 3.5 times better resolution. Ongoing and past remote sensing missions with hyperspectral and multispectral optical payloads like SeaWiFS, Aqua/MODIS, SNPP/VIIRS, ISS/HICO, Landsat8/OLI, DSCOVR/ EPIC, Sentinel-2/MSI, Sentinel-3/OLCI, COMS/GOCI *etc.*, facilitates a better ocean colour monitoring. The implementation of the Machine Learning algorithm on data obtained from these sensors boosts the efficiency of the model 10 times compared to the literature (*Fan et al., 2021*).

There exist several state-of-the-art technologies that have demonstrated the usage of remote sensing along with machine learning to study ocean surfaces as well as ocean beds (*Zhou et al., 2023*). For the surface studies, several works have attempted to detect several oceanographic parameters like chlorophyll-a, salinity, temperature, water movement dynamics, oil spills and many more (*Abou Samra, El-Gammal & Eissa, 2021*; *Kim et al., 2023*; *Prakash Tiwari, Adhikary & Banerjee, 2022*). Studies have also been performed to determine the correlation between oceanic phytoplankton and other oceanographic properties (*Li et al., 2023*; *Adhikary et al., 2021*). Remote sensing has also been used to study the seasonal variability of phytoplankton for about 50 years time scale but limited to only specific parts of the world (*Barton, Lozier & Williams, 2015*). Further, studies have also been performed to provide more advanced phytoplankton monitoring methods which can be built with ground-based remote sensing. However, this study is conducted on only two different lakes in China and that too for detecting blooms in real time and no means for future forecasting. The study (*Zhu et al., 2019*) shows that hyperspectral imagery with the help of transfer learning can be used for the detection or segmentation of phytoplankton but this study too couldn't analyse the dynamics of phytoplankton. Recent advancements in remote sensing combined with *in-situ* observations, and a reanalysis of data have been produced by the Marine Copernicus program to indicate the presence of various oceanographic parameters in different parts of the world for over 18 years (*Irazoqui Apecechea, Melet & Armaroli, 2023*). However, in the literature, there exist no studies to use such historically reanalyzed remote sensing data for a smart and in-depth study of global phytoplankton. This motivated us to conduct this experiment.

## MATERIALS AND METHODS

The entire experiment was conducted on a personal computer with 16 GB Random Access Memory (RAM), 4 GB NVIDIA GTX 1650ti Graphical Processing unit (GPU), 4 Core hyperthreaded intel i5 CPU and 1TB Non-Volatile Memory Express (NVMe) M.2 Solid

State Drive (SSD) was used. Linux operating system of kernel 5.11 was used for the experiment and all the operations were performed using Python 3.7 using various libraries like scikit-learn, matplotlib, numpy, pandas, seaborn and xarray.

## Data collection

For our experiment (the learning algorithms), the open-sourced dataset by the Marine Copernicus Program named Global Ocean Biogeochemistry Hindcast GLOBAL REANALYSIS BIO 001 029 monthly (*Perruche, 2018*), which contained biochemical records, and Global Reanalysis Phy 001 030 monthly, which contained physical records in NetCDF format. The resolution of the dataset was 4 km per pixel. This dataset contains the monthly-recorded data starting from 16th Jan 2000 till 16th Dec 2018. Altogether we have 206316 data points for the experiment. 80% of this data was used to train the model and the remaining 20% was used to validate the performances (*Prakash Tiwari, Adhikary & Banerjee, 2022*).

Among all these many features, surface $CO_2$, dissolved oxygen, nitrate, phosphate, dissolved silicate, pH, dissolved iron, ocean mixed layer thickness, surface temperature, sea floor temperature, northward velocity, eastward velocity, salinity and sea surface height were considered while building the model. On the other hand, the phytoplankton concentration was used as a target variable to train the model.

### Study area

To conduct this experiment, coordinate slices for multiple locations around the global oceans, were manually examined for time-dependent variations in phytoplankton concentrations. Based on substantial variation in multiple physical and chemical factors over time, six different locations (*i.e.*, Gulf of Mexico, Bay of Bengal, North Atlantic, North Pacific, South Atlantic and the Indian Ocean). Figure 1 depicts the portion of the Gulf of Mexico, the Bay of Bengal, the North Atlantic, the North Pacific, the South Atlantic and the Indian Ocean. However, depth remains constant in this study, *i.e.*, up to 0.5 m deep from the sea level. Based on this, a time-series data was created, as explained in the subsequent section. The descriptive statistics of the dataset are summarized in the table provided as Supplemental Files.

### Significance of each variable

Each one of the features used in the experiment has significant importance in the life cycle of oceanic phytoplankton. Surface $CO_2$ is an essential indicator as phytoplankton absorbs $CO_2$ depleting the concentration of the surface $CO_2$ (*Takao et al., 2020*). Dissolved oxygen is another essential parameter as phytoplankton are responsible for the world's 50% oxygen production (*Cai et al., 2022*). Nitrates and phosphates are important to forecast phytoplankton dynamics because growing phytoplankton requires them as nutrients and a continuous depletion in their concentration can indicate phytoplankton blooms (*Nindarwi, Samara & Santanumurti, 2021*). Diatoms, which is a type of phytoplankton form their frustules with the help of dissolved silicates and therefore this is another important parameter which is directly related to phytoplankton growth (*Xu et al., 2022*). Now, pH is a very important factor as phytoplankton best grows at pH levels of 7.5 to 8

(*Jia et al., 2020*). Dissolved iron is used by the phytoplankton as a micronutrient and increases its environmental stress tolerance (*Yuan et al., 2021*). Further, density ocean mixed layer thickness is an important parameter as with increasing thickness, light availability decreases which reduces phytoplankton metabolism (*Diaz et al., 2021*). A temperature of 18 deg–14 deg C is ideal for phytoplankton growth (*Fernández-González & Marañón, 2021*). Sea floor potential temperature is necessary for this study as a high difference between surface and sea floor potential temperature can indicate a survival difficulty for phytoplankton (*Chen et al., 2021*; *Zohary, Flaim & Sommer, 2021*). Northward or eastward velocity is important for the study because it helps in the dispersal of the phytoplankton, however, turbulent water streams can cause damage to the phytoplankton ecosystem (*Fai et al., 2023*). Sea surface height indicates the difference in distance between the mean sea surface and the reference ellipsoid (*Chen et al., 2021*). This parameter has been included in the study because it indicates the volume of the water from which the phytoplankton are measured.

## Data conversion to time series and preprocessing

The datasets obtained in the experiment are in a form analogous to an image where each pixel carries information about the magnitude of all biogeochemical features for the past 18 years for each consecutive day. Each pixel is associated with a coordinate. Information from each pixel was then extracted and converted to a time series set. Following that, we combined all of the pixelated time-series sets into a single time-series dataset. Following this step, we normalized the entire dataset to fit in the range of 0 and 1 by using the min-max scaler method.

Following this, data splits were performed in two stages. The first stage was where global data was combined, first 80% data from the total 18-year timescale was used to train the model and the remaining 20% was used to test the model. Later, the study was extended to train on all water bodies leaving one and then tests were performed on that remaining one. This was then repeated for all other water bodies. Subsequently, the predictions were then compared against both seen and unseen datasets to understand the fitness and eliminate any overfitting or underfitting issues.

The experiment was carried out on the Copernicus Hindcast dataset using different machine-learning algorithms, which are elaborated in the subsequent paragraphs. Figure 2 shows the prediction analysis work diagram for the proposed work.

## Regressors

To demonstrate and compare the applicability of different types of regressors, we conducted a test on four different tree-based algorithms as these are very versatile. These are Histogram-based Gradient Boosting Regressor (HGBR), Random Forest, Bagging and Extra Trees. The reason for these tree-based methods working best for this experiment is that the pre-processed data has many parameters which are dependent upon each other and for these situations, tree-based models work the best (*Park et al., 2019*). The random forest model built for this experiment contains 100 estimators which work based on mean

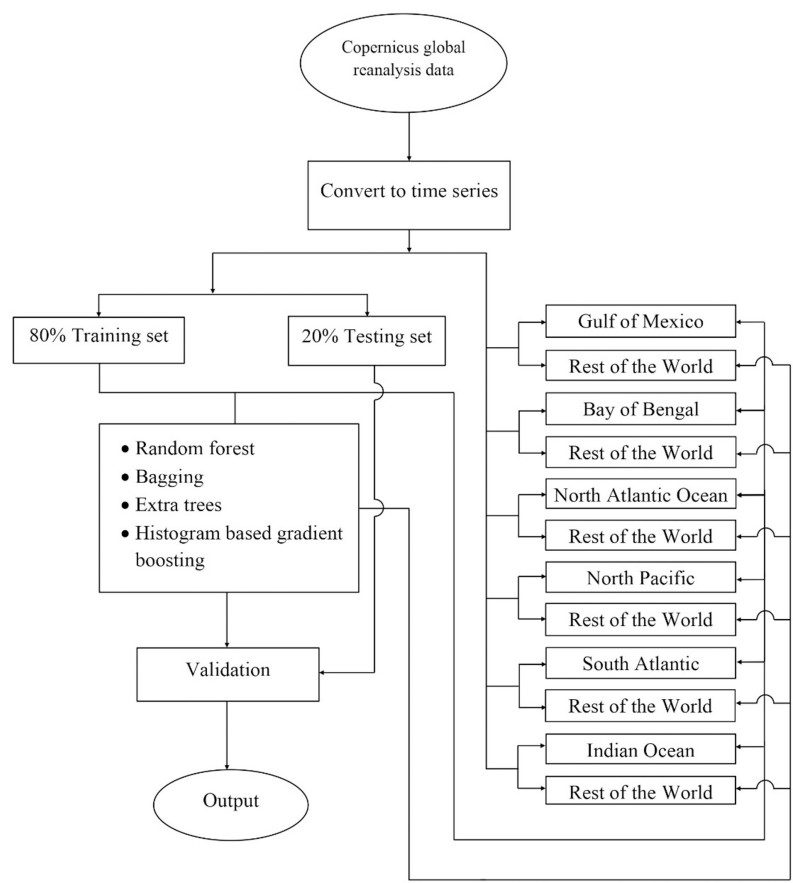

**Figure 2 The flowchart depicting the complete procedure for estimating phytoplankton concentrations in the present study.** There, the reanalyzed remote sensing data was obtained for six different locations around the world using the Copernicus program, later the data were converted into a time series, and following this the data splits were performed according to two different strategies i) 80% global data to train and rest 20% global data to test, ii) training set consisting all water bodies leaving one and later testing on that one which was left, and then finally machine learning was used to train the model. Later performances of the model have been validated.

squared error and minimal cost-complexity pruning parameter to be 0 (*Stock & Subramaniam, 2020*). The bagging regressor model was trained using 10 estimators (*Xu et al., 2023*). The extra tree was built to have an early stopping criterion for splitting a tree based on the threshold value of 2. The tree requires at least one sample to form a leaf node(*Wei et al., 2019*). The histogram-based gradient boosting model built for the experiment is tuned by least-square losses with a learning rate of 0.1 and has a maximum of 100 iterations. There is a maximum of 31 leaf nodes. There is a minimum of 20 samples per leaf node. 10% among the training data have been used for measuring the early stopping criteria. A maximum of 1e-7 absolute tolerance was used for comparing scores during early stopping. The early stopping was performed if the validation performance did not improve during the last 10 iterations (*Su et al., 2021*).

## RESULTS

Phytoplankton concentration at studied locations for different months revealed a typical surge in the northern hemisphere during March and April; on the other hand, phytoplankton concentration surged in the southern hemisphere during October and November. There is a significant natural decline in phytoplankton concentration about 75% by mass in the opposite hemisphere during peak months for another hemisphere. This phytoplankton life cycle has a number of consequences for the ecosystem, as excessive concentration could affect the ocean colour and too low concentration could possibly affect the food chain of various aquatic systems. When phytoplankton concentration declines, a similar decline in primary productivity occurs, as there is a large volume of phytoplankton available to contribute a significant amount of photosynthesis. When phytoplankton volume reaches its peak, the primary productivity also peaks and concentrations are dropped significantly. Figure 3 shows a low concentration of phytoplankton and Fig. 4 shows a higher concentration of phytoplankton. After observing the dataset, a strong relationship between the biogeochemical and physical features with the phytoplankton volumes can be deduced, implying that the natural phytoplankton life cycle occurs. These characteristics were used to estimate phytoplankton volumes in this experiment, and their performances are discussed in the following section.

### Phytoplankton estimation performances

The model has successfully estimated global phytoplankton concentration with a maximum of 0.963142 $R^2$ score and the performance summary is summarized in Table 1. From the experiment, when the model has been tested with the same data it was trained upon, it could be noted that the Extra Trees algorithm seems to be a perfect regressor having estimated with almost no error. However, this is not the case because the way this algorithm works is to first learn the data, *i.e.*, it predicts the same results when the same inputs are encountered but when there is a different input, it uses the average of closest values to predict the outcomes. So generally, the output of the Extra Trees could not be considered as overfitting as the prediction results on unseen data are good as well. To better understand the fitness, in the first test case, let us see the performances on the seen data. It can be observed that extra trees (3.02e-24) have the lowest mean squared error rates followed by random forest (0.000508), bagging (0.000771) and HGBR (0.005344). Extra trees have the best $R^2$ score of 1.0 followed by random forest being 0.994511, Bagging 0.991668 and HGBR being 0.942291 the lowest. The same pattern can be noticed for all other metrics as well. However, training the model, took the longest (331.683 s) with random forest as it requires building a large number of trees to train the model. Extra trees took 103.444 s, Bagging regressor took 33.538 and finally, HGBR was the fastest to train the model at 2.196 s. Following this, for the second test case, the testing was performed on unseen data. In this case, results however had a slightly lower performance but not to the point it could be considered overfitting. This time, the $R^2$ score was highest for extra trees (0.963142) as well followed by random forest (0.961293), bagging (0.956989) and HGBR (0.936631) being the lowest of all. From these two scenarios, it can be observed that the performances on the unseen data are slightly lower than the performances on the seen
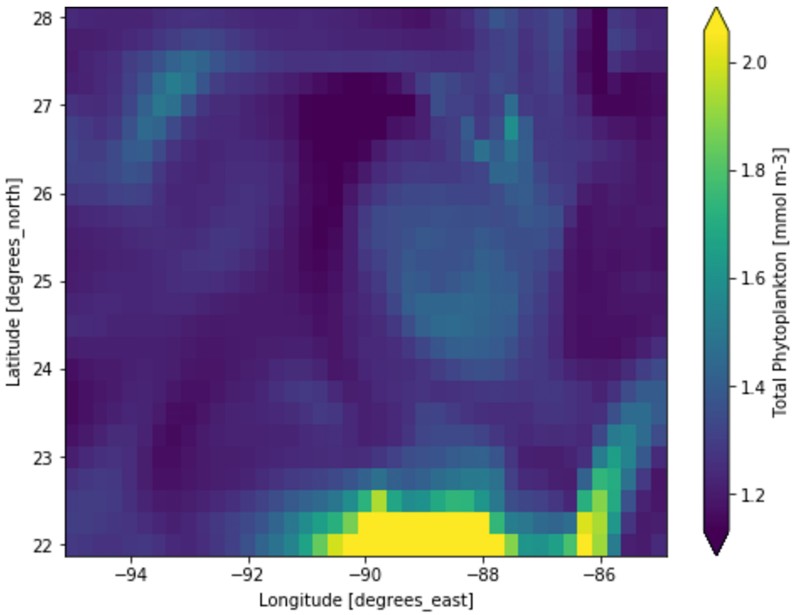

**Figure 3 A portion of the study location having low phytoplankton volumes.** Each pixelated block represents the aggregated value of phytoplankton concentration. The concentration of the phytoplankton in a particular pixel/region is indicated by the colour corresponding to the provided colour bar.

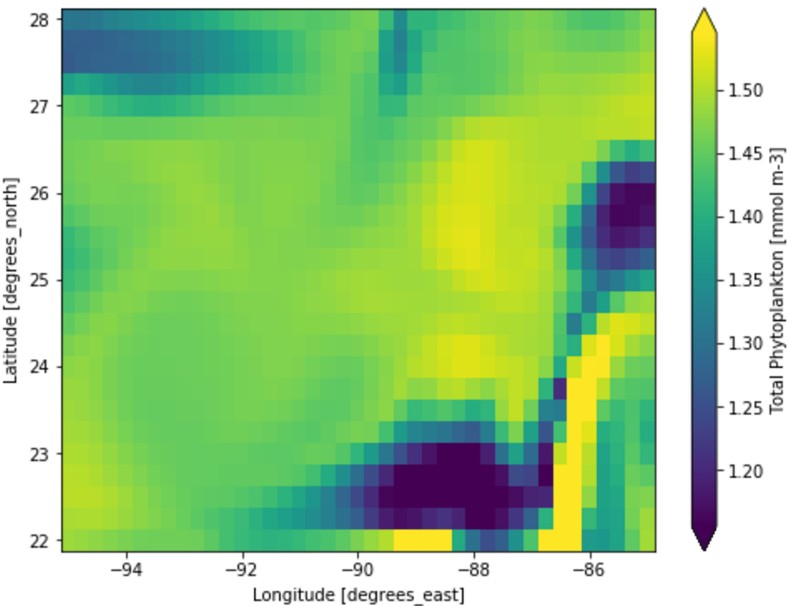

**Figure 4 A portion of the study location having high phytoplankton volumes.** Each pixelated block represents the aggregated value of phytoplankton concentration. The concentration of the phytoplankton in a particular pixel/region is indicated by the colour corresponding to the provided colour bar.

**Table 1 Regression performance for phytoplankton estimation in global waters.**

|  | Metric | Random forest | Bagging | Extra trees | HGBR |
|---|---|---|---|---|---|
| Train | Mean squared error | 0.000508 | 0.000771 | 3.02E−24 | 0.005344 |
|  | Median absolute error | 0.009533 | 0.010231 | 8.88E−16 | 0.038781 |
|  | $R^2$ score | 0.994511 | 0.991668 | 1.0 | 0.942291 |
|  | Explained variance | 0.994512 | 0.991668 | 1.0 | 0.942291 |
|  | Max error | 0.753581 | 0.885730 | 4.99E−10 | 1.574378 |
|  | Mean squared log error | 8.34E−05 | 0.000126 | 3.37E−25 | 0.000948 |
|  | Mean poisson deviance | 0.000349 | 0.000531 | 2.67E−18 | 0.003931 |
|  | Mean gamma deviance | 0.000268 | 0.000406 | −4.54E−17 | 0.003138 |
|  | Mean tweedie deviance | 0.000508 | 0.000771 | 3.02e-24 | 0.005344 |
|  | Train time (s) | 331.683 | 33.538 | 103.444 | 2.196 |
| Test | Mean squared error | 0.003565 | 0.003962 | 0.003395 | 0.005837 |
|  | Median absolute error | 0.026137 | 0.027924 | 0.025500 | 0.039536 |
|  | $R^2$ score | 0.961293 | 0.956989 | 0.963142 | 0.936631 |
|  | Explained variance | 0.961294 | 0.956990 | 0.963142 | 0.936631 |
|  | Max error | 1.396569 | 1.263875 | 1.077332 | 1.491765 |
|  | Mean squared log error | 0.000596 | 0.000673 | 0.000570 | 0.001011 |
|  | Mean poisson deviance | 0.002495 | 0.002811 | 0.002383 | 0.004215 |
|  | Mean gamma deviance | 0.001919 | 0.002175 | 0.001835 | 0.003323 |
|  | Mean tweedie deviance | 0.003565 | 0.003962 | 0.003395 | 0.005837 |
|  | Test time (s) | 2.211391 | 0.232537 | 2.426239 | 0.089830 |

data, although both display promising results. This confirms that there exists no case of overfitting.

The sequence is the same as a prediction on training data. The time it took to produce results is the fastest for HGBR (0.089830 s) followed by bagging (0.232537 s), random forest (2.21391 s) and extra tree (2.426239 s) the slowest. Therefore although the overall accuracy of Extra Trees is higher than the remaining algorithms, the training and testing time is very slow indicating a higher amount of resource consumption hence in general, Bagging could be the best suitable model both in terms of accuracy and speed. This time difference is considerably high considering deployment scenarios where the model is used as a server handling thousands of concurrent requests and in this case, a low latency response is necessary. The estimations predicted by the models and the original have been plotted with heatmaps corresponding to the concentration of the points have been shown in Fig. 5. Figure 5A shows the performance for prediction with global data. It can be observed that the prediction for this was the most accurate compared to all other locations this is because this location had data from all parts of the world. Likewise, Fig. 2 represents the prediction for the North Pacific Ocean when trained with water from the remaining parts of the world. After the globally trained model, this model had the lowest prediction deviation for all the algorithms. This clearly signifies that the water properties of this location have the highest similarity with the remaining part of the world in terms of the

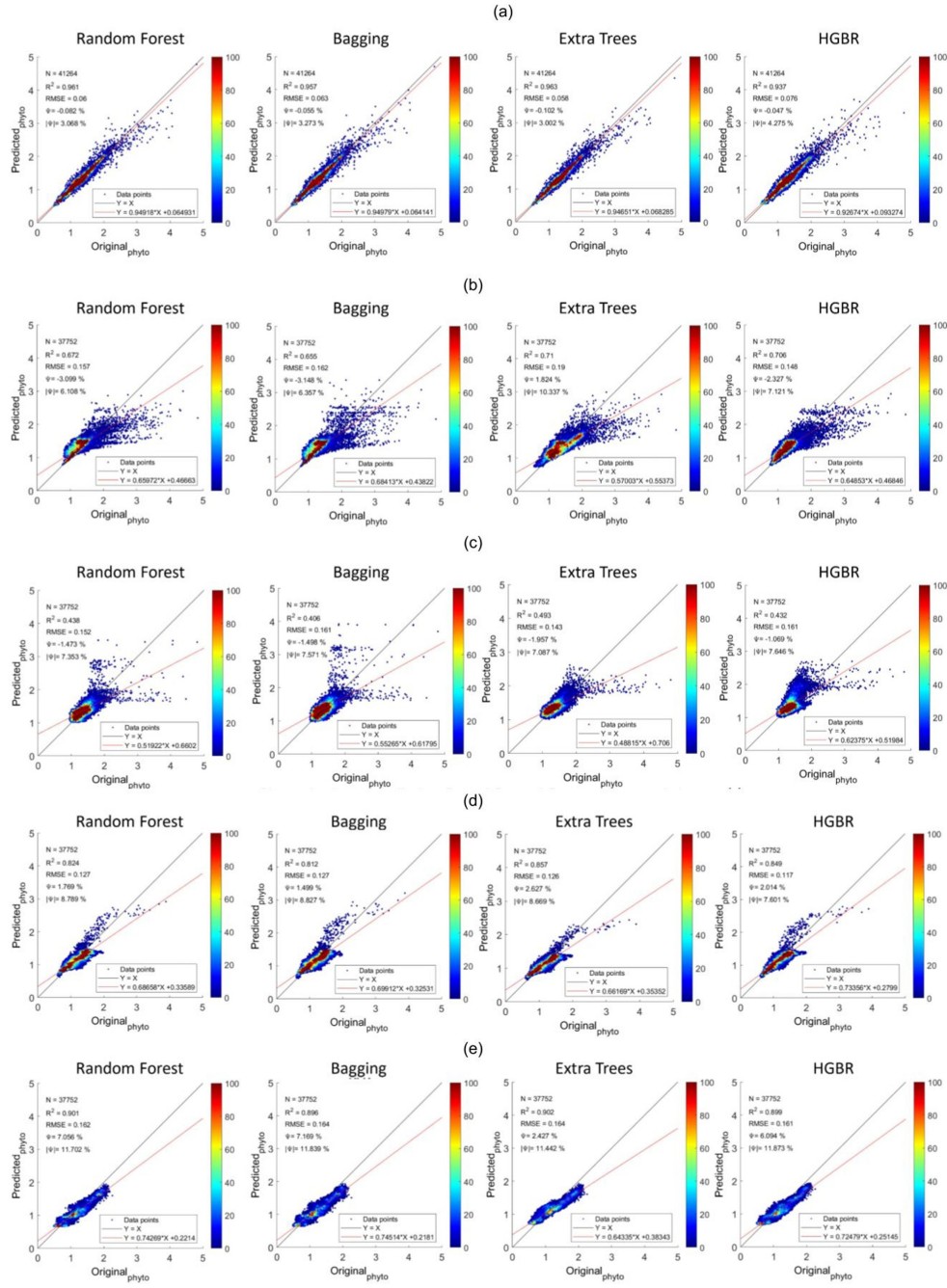

**Figure 5 Comparisons of original phytoplankton concentration *vs.* all the modelled phytoplankton concentration obtained from random forest, bagging, extra trees, and HGBR, for all locations combined and one *vs.* rest locations.** The points along the x-axis show the original concentration of phytoplankton and the y-axis shows the predicted concentration of phytoplankton. The solid black and red lines represent the one-to-one line and regression line, respectively. A smaller angle between the lines with a small distance between the lines (when parallel) represents a better fit. The colour bar shows the concentration of the points in the range. The sub-graphs shown represents modeling performances based on data obtained from: (A) World-wide (B) Gulf of Mexico (C) Bay of Bengal (D) North Atlantic (E) North Pacific (F) South Atlantic (G) Indian Ocean.

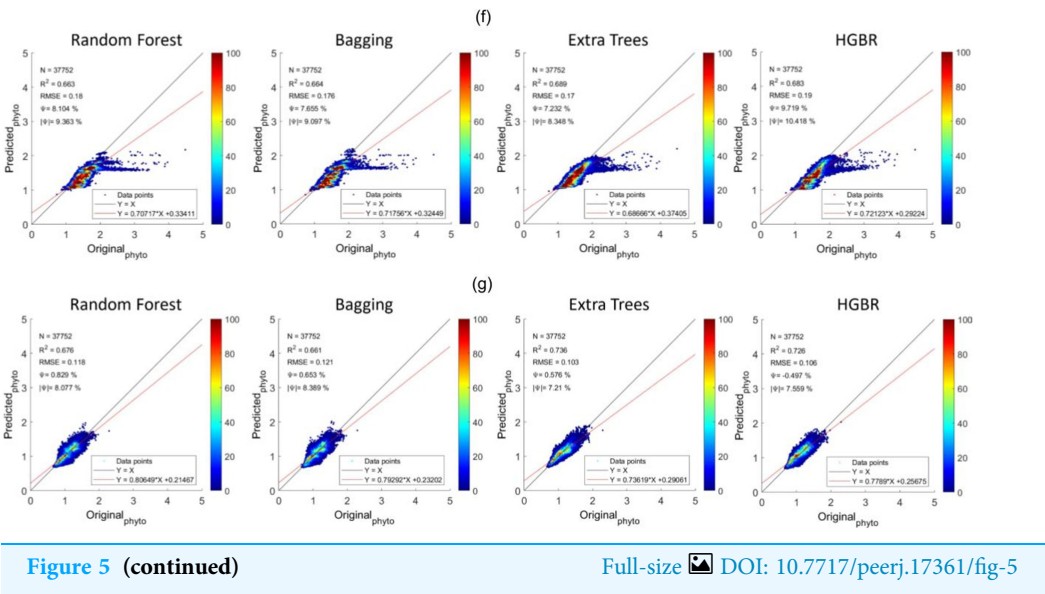

**Figure 5 (continued)**               

earlier-mentioned physiochemical features. Similarly, the amount of performance deviation from all the locations can be used to identify the amount of variations in the physiochemical properties of these study areas.

## Addressing underfitting and global variations of biogeochemical features

The waters worldwide have a different composition of biogeochemical features and they have a different impact on the lifecycle of phytoplankton with different evolutionary traits. Therefore it is important to understand the effects of these variations in automated estimations of phytoplankton. Different areas could have different composition patterns and when this pattern is not known to the model, it could generate degraded results. Therefore the experiment has been extended to estimate phytoplankton volumes of locations that are not present in training data and test the model in order to understand which areas have high variations and need more attention while developing the best fit for the model.

From the aforementioned tables, the North Pacific Ocean (Table 2), North Atlantic Ocean (Table 3) and South Atlantic Ocean (Table 4) all have reasonable $R^2$ scores around 0.8, indicating that these three locations have high similarity with the remaining oceans in the world (Table 1). $R^2$ score for the Indian Ocean (Table 5) is not as high as the others, but still within an acceptable $R^2$ score range of roughly 0.65–0.7, showing a somewhat different biogeochemical feature pattern and its relationship with phytoplankton. The Gulf of Mexico (Table 6) and the Bay of Bengal (Table 7) have quite low $R^2$ scores in the range of 0.4 to 0.5, indicating higher variations of biogeochemical properties and corresponding phytoplanktons. With high variations associated with specific locations, estimating phytoplankton in certain locations is difficult, resulting in an underfitting problem. Therefore to improve the estimation for these regions, some data from these locations

**Table 2 Regression performance for North Pacific _vs_ rest of the world.**

|  | Metric | Random forest | Bagging | Extra trees | HGBR |
|---|---|---|---|---|---|
| Train | Mean squared error | 0.000550 | 0.000844 | 1.39E−30 | 0.005588 |
|  | Median absolute error | 0.009817 | 0.010455 | 8.88E−16 | 0.039067 |
|  | $R^2$ score | 0.993147 | 0.989483 | 1.0 | 0.930399 |
|  | Explained variance | 0.993147 | 0.989483 | 1.0 | 0.930399 |
|  | Max error | 0.780050 | 1.583320 | 1.24E−14 | 1.613996 |
|  | Mean squared log error | 8.78e−05 | 0.000134 | 2.41E−31 | 0.000973 |
|  | Mean poisson deviance | 0.000369 | 0.000566 | 2.10E−18 | 0.004041 |
|  | Mean gamma deviance | 0.000276 | 0.000420 | −4.64E−17 | 0.003144 |
|  | Mean tweedie deviance | 0.000550 | 0.000844 | 1.39E−30 | 0.005588 |
|  | Train time (s) | 332.995 | 33.121 | 93.910 | 4.218 |
| Test | Mean squared error | 0.025677 | 0.026768 | 0.026934 | 0.026008 |
|  | Median absolute error | 0.113882 | 0.116661 | 0.124013 | 0.120700 |
|  | $R^2$ score | 0.819864 | 0.812213 | 0.811046 | 0.817542 |
|  | Explained variance | 0.874255 | 0.870377 | 0.828022 | 0.865166 |
|  | Max error | 0.645404 | 0.752089 | 0.654208 | 0.545281 |
|  | Mean squared log error | 0.004973 | 0.005199 | 0.005137 | 0.005100 |
|  | Mean poisson deviance | 0.020916 | 0.021870 | 0.021323 | 0.021372 |
|  | Mean gamma deviance | 0.018336 | 0.019243 | 0.018447 | 0.018981 |
|  | Mean tweedie deviance | 0.025677 | 0.026768 | 0.026934 | 0.026008 |
|  | Test time (s) | 0.544131 | 0.069268 | 0.521268 | 0.074275 |

**Table 3 Regression performance for North Atlantic _vs_ rest of the world.**

|  | Metric | Random forest | Bagging | Extra trees | HGBR |
|---|---|---|---|---|---|
| Train | Mean squared error | 0.000540 | 0.000797 | 1.47E−30 | 0.005366 |
|  | Median absolute error | 0.009622 | 0.010237 | 8.88E−16 | 0.038228 |
|  | $R^2$ score | 0.994164 | 0.991390 | 1.0 | 0.942082 |
|  | Explained variance | 0.994164 | 0.991390 | 1.0 | 0.942082 |
|  | Max error | 0.843920 | 0.879325 | 1.24E−14 | 1.913743 |
|  | Mean squared log error | 8.66e−05 | 0.000129 | 2.48E−31 | 0.000927 |
|  | Mean poisson deviance | 0.000364 | 0.000542 | 2.39E−18 | 0.003860 |
|  | Mean gamma deviance | 0.000273 | 0.000409 | −4.55E−17 | 0.003028 |
|  | Mean tweedie deviance | 0.000540 | 0.000797 | 1.47E−30 | 0.005366 |
|  | Train time (s) | 339.787 | 33.872 | 89.379 | 3.595 |
| Test | Mean squared error | 0.015781 | 0.016177 | 0.015921 | 0.013633 |
|  | Median absolute error | 0.081522 | 0.083380 | 0.080002 | 0.067542 |
|  | $R^2$ score | 0.793842 | 0.788661 | 0.792010 | 0.821896 |
|  | Explained variance | 0.802206 | 0.796368 | 0.812456 | 0.833406 |
|  | Max error | 1.103184 | 0.725113 | 1.575166 | 1.176767 |
|  | Mean squared log error | 0.003187 | 0.003281 | 0.003122 | 0.002609 |

(Continued)

| | Metric | Random forest | Bagging | Extra trees | HGBR |
|---|---|---|---|---|---|
| | Mean poisson deviance | 0.013135 | 0.013503 | 0.012982 | 0.010815 |
| | Mean gamma deviance | 0.011459 | 0.011816 | 0.011066 | 0.008956 |
| | Mean tweedie deviance | 0.015781 | 0.016177 | 0.015921 | 0.013633 |
| | Test time (s) | 0.809874 | 0.094937 | 0.894909 | 0.089111 |

**Table 4 Regression performance for South Atlantic *vs* rest of the world.**

| | Metric | Random forest | Bagging | Extra trees | HGBR |
|---|---|---|---|---|---|
| Train | Mean squared error | 0.000550 | 0.000844 | 1.39E−30 | 0.005588 |
| | Median absolute error | 0.009817 | 0.010455 | 8.88E−16 | 0.039067 |
| | $R^2$ Score | 0.993147 | 0.989483 | 1.0 | 0.930399 |
| | Explained variance | 0.993147 | 0.989483 | 1.0 | 0.930399 |
| | Max error | 0.780050 | 1.583320 | 1.24E−14 | 1.613996 |
| | Mean squared log error | 8.78e-05 | 0.000134 | 2.41E−31 | 0.000973 |
| | Mean poisson deviance | 0.000369 | 0.000566 | 2.10E−18 | 0.004041 |
| | Mean gamma deviance | 0.000276 | 0.000420 | −4.64E−17 | 0.003144 |
| | Mean tweedie deviance | 0.000550 | 0.000844 | 1.39E−30 | 0.005588 |
| | Train time (s) | 332.995 | 33.121 | 93.910 | 4.218 |
| Test | Mean squared error | 0.025677 | 0.026768 | 0.026934 | 0.026008 |
| | Median absolute error | 0.113882 | 0.116661 | 0.124013 | 0.120700 |
| | $R^2$ score | 0.819864 | 0.812213 | 0.811046 | 0.817542 |
| | Explained variance | 0.874255 | 0.870377 | 0.828022 | 0.865166 |
| | Max error | 0.645404 | 0.752089 | 0.654208 | 0.545281 |
| | Mean squared log error | 0.004973 | 0.005199 | 0.005137 | 0.005100 |
| | Mean poisson deviance | 0.020916 | 0.021870 | 0.021323 | 0.021372 |
| | Mean gamma deviance | 0.018336 | 0.019243 | 0.018447 | 0.018981 |
| | Mean tweedie deviance | 0.025677 | 0.026768 | 0.026934 | 0.026008 |
| | Test time (s) | 0.544131 | 0.069268 | 0.521268 | 0.074275 |

should be incorporated while training the model to facilitate the model to learn properties associated with these regions. This enhances the overall efficiency of phytoplankton estimation in global waters.

## Regression for seasonal variability

Later, the investigation has been extended to forecast the seasonal variability of phytoplankton. Data of various seasons for this purpose have been collected as an arbitrary day from the 2nd week of April, July, October and January as spring, summer, fall and winter respectively for the northern hemisphere and fall, winter, spring and summer respectively for the southern hemisphere. After performing the regression of the data obtained from all over the world, the performance summary has been recorded in Table 8.

**Table 5 Regression performance for Indian Ocean vs rest of the world.**

|  | Metric | Random forest | Bagging | Extra trees | HGBR |
|---|---|---|---|---|---|
| Train | Mean squared error | 0.000483 | 0.000719 | 1.49E−30 | 0.004953 |
|  | Median absolute error | 0.008625 | 0.009333 | 8.88E−16 | 0.036254 |
|  | $R^2$ score | 0.995086 | 0.992686 | 1.0 | 0.949648 |
|  | Explained variance | 0.995086 | 0.992686 | 1.0 | 0.949648 |
|  | Max error | 1.000382 | 0.940252 | 1.24E−14 | 1.585642 |
|  | Mean squared log error | 7.31e-05 | 0.000110 | 2.48E−31 | 0.000841 |
|  | Mean poisson deviance | 0.000310 | 0.000468 | 2.28E−18 | 0.003511 |
|  | Mean gamma deviance | 0.000226 | 0.000342 | −4.65E−17 | 0.002728 |
|  | Mean tweedie deviance | 0.000483 | 0.000719 | 1.49E−30 | 0.004953 |
|  | Train time (s) | 333.271 | 33.197 | 94.616 | 2.690 |
| Test | Mean squared error | 0.013952 | 0.014648 | 0.010607 | 0.011180 |
|  | Median absolute error | 0.068951 | 0.074891 | 0.067989 | 0.072918 |
|  | $R^2$ score | 0.651281 | 0.633906 | 0.734892 | 0.720565 |
|  | Explained variance | 0.651734 | 0.633955 | 0.736032 | 0.721982 |
|  | Max error | 0.564515 | 0.614417 | 0.536458 | 0.466304 |
|  | Mean squared log error | 0.003003 | 0.003157 | 0.002225 | 0.002411 |
|  | Mean poisson deviance | 0.012201 | 0.012809 | 0.009058 | 0.009761 |
|  | Mean gamma deviance | 0.010946 | 0.011487 | 0.007882 | 0.008729 |
|  | Mean tweedie deviance | 0.013952 | 0.014648 | 0.010607 | 0.011180 |
|  | Test time (s) | 0.771438 | 0.093410 | 0.926036 | 0.105368 |

**Table 6 Regression performance for Gulf of Mexico vs rest of the world.**

|  | Metric | Random forest | Bagging | Extra trees | HGBR |
|---|---|---|---|---|---|
| Train | Mean squared error | 0.000382 | 0.000580 | 1.36E−30 | 0.004523 |
|  | Median absolute error | 0.008875 | 0.009519 | 8.88E−16 | 0.036596 |
|  | $R^2$ score | 0.995809 | 0.993642 | 1.0 | 0.950445 |
|  | Explained variance | 0.995809 | 0.993642 | 1.0 | 0.950445 |
|  | Max error | 0.612501 | 0.662656 | 1.24E−14 | 1.729356 |
|  | Mean squared log error | 6.89e-05 | 0.000104 | 2.35E−31 | 0.000845 |
|  | Mean poisson deviance | 0.000285 | 0.000433 | 2.79E−18 | 0.003482 |
|  | Mean gamma deviance | 0.000230 | 0.000349 | −4.54E−17 | 0.002877 |
|  | Mean tweedie deviance | 0.000382 | 0.000580 | 1.36E−30 | 0.004523 |
|  | Train time (s) | 400.385 | 38.076 | 102.926 | 3.532 |
| Test | Mean squared error | 0.053334 | 0.055418 | 0.041754 | 0.040747 |
|  | Median absolute error | 0.138261 | 0.141163 | 0.107952 | 0.108073 |
|  | $R^2$ score | 0.479236 | 0.458879 | 0.592300 | 0.602135 |
|  | Explained variance | 0.549716 | 0.535834 | 0.626026 | 0.630668 |
|  | Max error | 2.730470 | 2.643686 | 2.994876 | 2.540424 |

(Continued)

| Metric | Random forest | Bagging | Extra trees | HGBR |
|---|---|---|---|---|
| Mean squared log error | 0.007989 | 0.008365 | 0.005791 | 0.005754 |
| Mean poisson deviance | 0.034055 | 0.035548 | 0.025349 | 0.025048 |
| Mean gamma deviance | 0.023269 | 0.024310 | 0.016734 | 0.016660 |
| Mean tweedie deviance | 0.053334 | 0.055418 | 0.041754 | 0.040747 |
| Test time (s) | 0.383338 | 0.044700 | 0.428533 | 0.043013 |

**Table 7 Regression performance for Bay of Bengal *vs* rest of the world.**

| | Metric | Random forest | Bagging | Extra trees | HGBR |
|---|---|---|---|---|---|
| Train | Mean squared error | 0.000398 | 0.000619 | 1.36E−30 | 0.004441 |
| | Median absolute error | 0.008311 | 0.008948 | 8.88E−16 | 0.035103 |
| | $R^2$ score | 0.996153 | 0.994025 | 1.0 | 0.957152 |
| | Explained variance | 0.996153 | 0.994025 | 1.0 | 0.957152 |
| | Max error | 0.820872 | 0.870304 | 1.06E−14 | 1.635578 |
| | Mean squared log error | 6.70e-05 | 0.000103 | 2.31E−31 | 0.000821 |
| | Mean poisson deviance | 0.000280 | 0.000431 | 3.46E−18 | 0.003390 |
| | Mean gamma deviance | 0.000223 | 0.000339 | −4.46E−17 | 0.002814 |
| | Mean tweedie deviance | 0.000398 | 0.000619 | 1.36E−30 | 0.004441 |
| | Train time (s) | 335.91 | 32.82 | 86.35 | 2.23 |
| Test | Mean squared error | 0.023537 | 0.025910 | 0.020538 | 0.025942 |
| | Median absolute error | 0.080000 | 0.081830 | 0.078491 | 0.079452 |
| | $R^2$ score | 0.398746 | 0.338133 | 0.475358 | 0.337315 |
| | Explained variance | 0.413666 | 0.352789 | 0.493099 | 0.347794 |
| | Max error | 4.256334 | 4.202001 | 4.055531 | 4.000856 |
| | Mean squared log error | 0.003358 | 0.003611 | 0.002976 | 0.003687 |
| | Mean poisson deviance | 0.014542 | 0.015622 | 0.012900 | 0.015831 |
| | Mean gamma deviance | 0.009806 | 0.010383 | 0.008821 | 0.010386 |
| | Mean tweedie deviance | 0.023537 | 0.025910 | 0.020538 | 0.025942 |
| | Test time (s) | 0.757761 | 0.093414 | 0.815170 | 0.147152 |

Here it can be clearly observed that the maximum $R^2$ score for regression was found from bagging regressor for winter data which is 0.989. Bagging regressor also estimated phytoplankton accurately during summer with $R^2$ score of 0.987. This high accuracy for summer and winter is because of the fact that during this time, the growth of depletion of phytoplankton comes to a stable rate. Highest difficulty was found during the fall as the $R^2$ score for all the regressors was found to be between 0.922 to 0.966. The reason for lower accuracy during fall was because of higher water frequency and intensity of storms and also with lowering temperature. However, phytoplankton can still bloom due to stratification and deepening of mixed layer which elevates nutrients from the ocean depths reaches the

**Table 8 Global phytoplankton dynamics regression for seasonal variability.**

| Season | Metric | Random forest | Bagging | Extra trees | HGBR |
|--------|--------|---------------|---------|-------------|------|
| Spring | Mean squared error | 0.003658 | 0.004063 | 0.003344 | 0.005819 |
| | Median absolute error | 0.024961 | 0.027586 | 0.025951 | 0.040845 |
| | $R^2$ score | 0.965811 | 0.937275 | 0.944842 | 0.967446 |
| | Explained variance | 0.954757 | 0.977374 | 0.997334 | 0.963606 |
| | Max error | 1.335260 | 1.220019 | 1.065912 | 1.555613 |
| | Mean squared log error | 0.000568 | 0.000641 | 0.000594 | 0.000968 |
| | Mean poisson deviance | 0.002455 | 0.002882 | 0.002403 | 0.004143 |
| | Mean gamma deviance | 0.002001 | 0.002121 | 0.001812 | 0.003476 |
| | Mean tweedie deviance | 0.003598 | 0.004012 | 0.003380 | 0.005932 |
| | Test time (s) | 2.220237 | 0.226235 | 2.394213 | 0.091788 |
| Summer | Mean squared error | 0.003549 | 0.003935 | 0.003295 | 0.005683 |
| | Median absolute error | 0.025669 | 0.028513 | 0.024404 | 0.038631 |
| | $R^2$ score | 0.972348 | 0.987134 | 0.952258 | 0.967446 |
| | Explained variance | 0.972349 | 0.923304 | 0.995985 | 0.897012 |
| | Max error | 1.406904 | 1.310006 | 1.083257 | 1.490870 |
| | Mean squared log error | 0.000621 | 0.000683 | 0.000576 | 0.001052 |
| | Mean poisson deviance | 0.002435 | 0.002904 | 0.002372 | 0.004295 |
| | Mean gamma deviance | 0.001877 | 0.002170 | 0.001864 | 0.003440 |
| | Mean tweedie deviance | 0.003705 | 0.003884 | 0.003475 | 0.005806 |
| | Test time (s) | 2.214708 | 0.243513 | 2.443223 | 0.094106 |
| Fall | Mean squared error | 0.003669 | 0.003798 | 0.003255 | 0.006076 |
| | Median absolute error | 0.026417 | 0.028494 | 0.024521 | 0.040920 |
| | $R^2$ score | 0.922553 | 0.920049 | 0.942916 | 0.966510 |
| | Explained variance | 0.996189 | 0.951152 | 0.969980 | 0.932229 |
| | Max error | 1.413607 | 1.297494 | 1.036932 | 1.425083 |
| | Mean squared log error | 0.000614 | 0.000667 | 0.000593 | 0.001024 |
| | Mean poisson deviance | 0.002542 | 0.002884 | 0.002426 | 0.004109 |
| | Mean gamma deviance | 0.001927 | 0.002259 | 0.001792 | 0.003172 |
| | Mean tweedie deviance | 0.003693 | 0.003949 | 0.003353 | 0.005545 |
| | Test time (s) | 2.213381 | 0.227328 | 2.321668 | 0.089210 |
| Winter | Mean squared error | 0.003703 | 0.003928 | 0.003357 | 0.005849 |
| | Median absolute error | 0.024898 | 0.027801 | 0.024233 | 0.039512 |
| | $R^2$ score | 0.968695 | 0.988857 | 0.924038 | 0.903568 |
| | Explained variance | 0.917459 | 0.963976 | 0.972388 | 0.897105 |
| | Max error | 1.384838 | 1.290290 | 1.059556 | 1.431349 |
| | Mean squared log error | 0.000573 | 0.000706 | 0.000598 | 0.001000 |
| | Mean poisson deviance | 0.002547 | 0.002776 | 0.002292 | 0.004150 |
| | Mean gamma deviance | 0.001843 | 0.002102 | 0.001786 | 0.003347 |
| | Mean tweedie deviance | 0.003444 | 0.003952 | 0.003318 | 0.005605 |
| | Test time (s) | 2.249427 | 0.234769 | 2.513584 | 0.090091 |

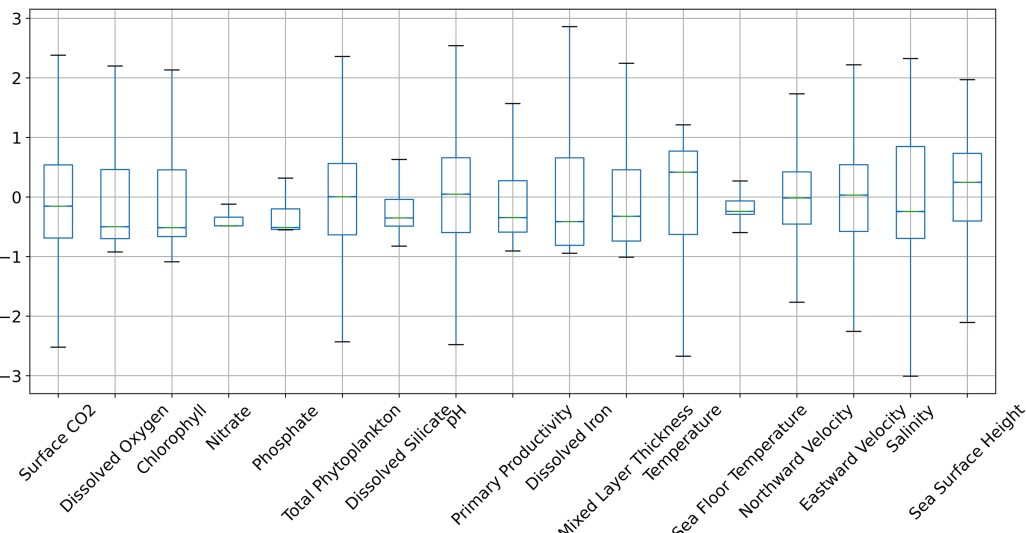

**Figure 6 The distribution of the biogeochemical features for all study locations combined, and standardized with a mean 0 and unit standard deviation.**

surface which can result in sudden blooms of phytoplankton. This behaviour is difficult to estimate which in turn causes a reduced accuracy for the estimation.

## DISCUSSION

The study surpasses the prior works in multiple aspects. To our knowledge, this is the first work to monitor the dynamics of marine phytoplankton which has studied more regions compared to any prior works (*Barton, Lozier & Williams, 2015*). Earlier works have been conducted to establish the dependency of physiochemical features with phytoplankton, but the proposed work is the first one to perform time series analysis of this property at a global scale (*Adhikary et al., 2021*). Further, prior works have been conducted for estimation of many physiochemical features but the proposed work is the only one which performed a time series analysis of these properties based on 18 years of data (*Prakash Tiwari, Adhikary & Banerjee, 2022*).

Figures 6 and 7 show the distribution of the biogeochemical features of waters from all study locations. Observing these plots, it can be noticed that the North Pacific Ocean has a wide spectrum of chlorophyll distribution but other regions have spiked toward 0–0.2 scale. North Pacific Ocean and South Atlantic Ocean have considerably large distributions of dissolved nitrates quantities but both Gulf of Mexico and Bay of Bengal have spikes toward the lower end of the spectrum and this could be a good indicator for the difference in regression performance in these regions. A similar pattern is also visible for dissolved phosphates. The pH of the Bay of Bengal has spiked toward a similar region whereas other areas have a tapered distribution and this could be an indicator of poor regression performance at the waters of the Bay of Bengal.

There are several biases and points which can cause errors in the result in the long run and which need to be fixed in subsequent experiments. One important point is that each pixel or data unit used for the experiment corresponds to 2 KM resolution averaging the

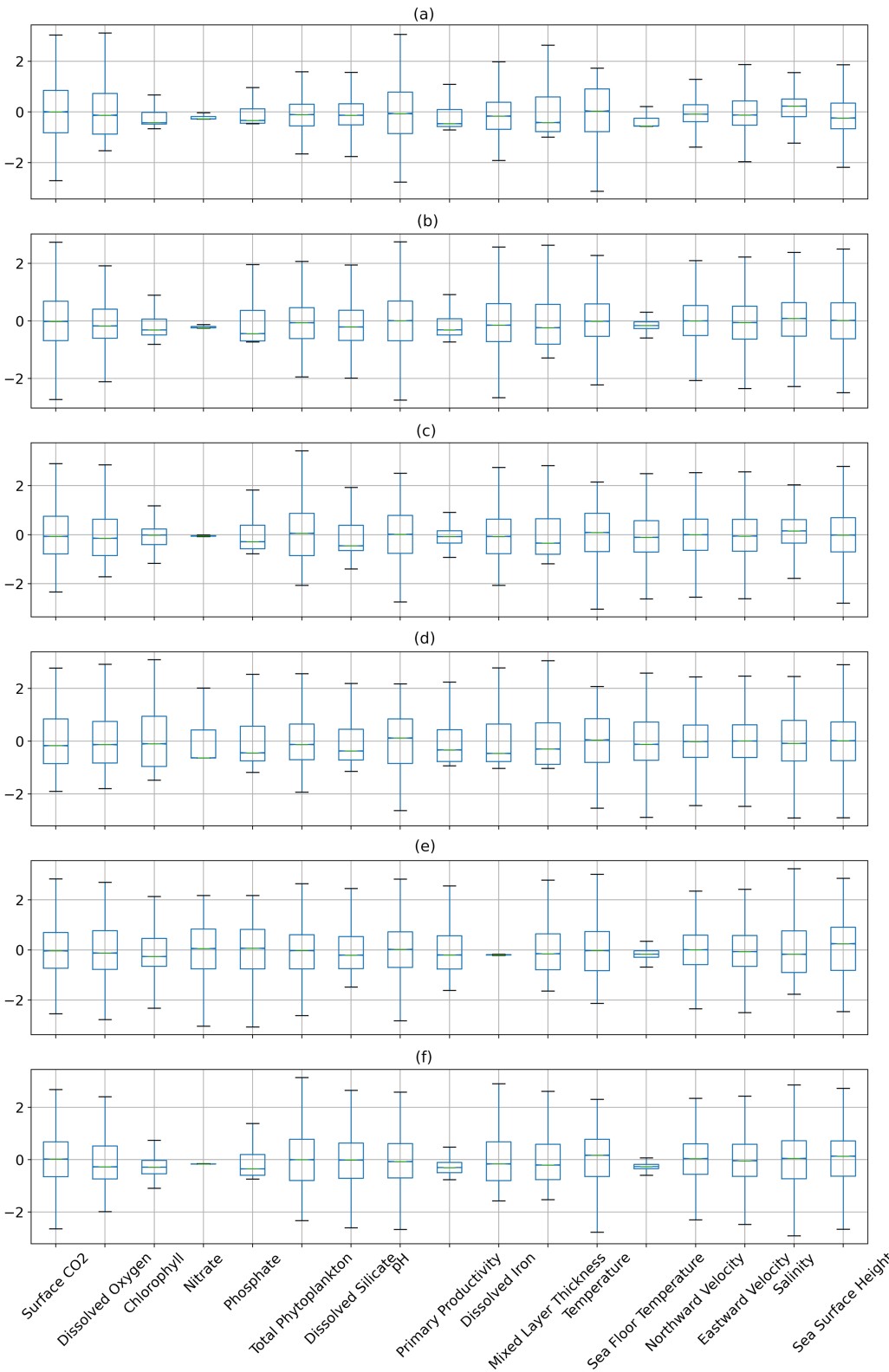

**Figure 7 The distribution of the biogeochemical features for all study locations separately, and standardized with a mean 0 and unit standard deviation.**

data of the entire 2 KM for one point. This covers a significantly large area and therefore minor variations within these regions are not recorded which can cause inconsistency while measuring data from a finer spot.

## CONCLUSION AND FUTURE WORKS

Phytoplankton concentrations in the Gulf of Mexico, Bay of Bengal, North Atlantic, North Pacific, South Atlantic, and Indian Ocean were predicted and forecasted using the biochemical and physical properties of these marine waters.

The extra trees regressor was found to perform best in this study, with an $R^2$ score of 0.963142, but the training and prediction times were long, at 103.444 and 2.426239 s, respectively. Following this, the bagging regressor has an $R^2$ score of 0.956989 and trains and predicts in 33.538 and 0.232537 s, respectively. Later, the trained model was tested on waters from six different parts of the world to investigate the variation in biogeochemical feature distribution worldwide. This has revealed that, while most parts of the world have a similar distribution of biogeochemical features, only a few regions vary significantly. Underfitting was avoided by incorporating data from waters with diverse biogeochemical distributions. Finally, regression has been tested to predict seasonal variability of global phytoplankton dynamics which has revealed that it is easiest to predict phytoplankton dynamics during summer and winter with $R^2$ scores of up to 0.987 and 0.988 respectively while it is difficult to predict the dynamics during spring and fall which have achieved a maximum $R^2$ score of 0.967 and 0.966 respectively.

Adding more independent features and locations with longer timeframes to the training set could further enhance the accuracy and reliability of the models. The machine learning models used in this study could be further enhanced by optimizing the hyper-parameters using different kinds of metaheuristics such as genetic algorithm, particle swarm optimization, gravitational search algorithm, and pathfinder algorithm.

## ACKNOWLEDGEMENTS

The authors would like to thank Copernicus Marine Service for providing the datasets that have been used to conduct present study. The authors would also like to thank Spiraldevs Automation Industries Pvt. Ltd., the Gyanam Foundation, Wingbotics LLP, King Fahd University of Petroleum & Minerals, Saudi Arabia and Aalborg University for the encouragement to conduct the study.

### Funding
The authors received no funding for this work.

### Competing Interests
Subhrangshu Adhikary is employed by Spiraldevs Automation Industries Pvt. Ltd. and Saikat Banerjee is employed by Wingbiotics.

## Author Contributions

- Subhrangshu Adhikary conceived and designed the experiments, performed the experiments, analyzed the data, prepared figures and/or tables, authored or reviewed drafts of the article, and approved the final draft.
- Surya Prakash Tiwari conceived and designed the experiments, performed the experiments, analyzed the data, prepared figures and/or tables, authored or reviewed drafts of the article, and approved the final draft.
- Saikat Banerjee conceived and designed the experiments, performed the experiments, analyzed the data, prepared figures and/or tables, authored or reviewed drafts of the article, and approved the final draft.
- Ashutosh Dhar Dwivedi conceived and designed the experiments, performed the experiments, analyzed the data, prepared figures and/or tables, authored or reviewed drafts of the article, and approved the final draft.
- Syed Masiur Rahman conceived and designed the experiments, performed the experiments, analyzed the data, prepared figures and/or tables, authored or reviewed drafts of the article, and approved the final draft.

## Data Availability

The code is available at Zenodo: Subhrangshu Adhikary. (2024). subhrangshu/ocm-regression: v1.0.0.2 (v1.0.0.2). Zenodo. https://doi.org/10.5281/zenodo.10500422.

## Supplemental Information

Supplemental information for this article can be found online at http://dx.doi.org/10.7717/peerj.17361#supplemental-information.

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
