# Peer review of "Global marine phytoplankton dynamics analysis with machine learning and reanalyzed remote sensing"

_PeerJ, doi:10.7717/peerj.17361_

## Round 0.1 · original submission · Major Revisions

1. Polish the title, more precise one is needed.

2. The authors should make a discussion about why this research is important in the introduction part.

3. The authors should provide a brief introduction regarding the coming sections at the end of the introduction part.

4. The authors need to provide some research background not only about phytoplankton, but also machine learning, or AI related. A lot more details about emote sensing, how you obtained the data etc are needed

5. The title “2. Methodology” may need to be changed to “2. Research methods” instead.

6. “There are two types of supervised machine learning: classification and regression.” Here the authors may make a brief introduction.

7. “Four different types of regressors were considered namely Random Forest Regressor, Bagging Regressor, Extra Trees Regressor…” Here the authors may also briefly introduce them with references.

8. “The time it took to produce results is 212

9. the fastest for HGBR (0.089830s) followed by Bagging (0.232537s), Random Forest (2.21391s) and Extra Tree (2.426239) the slowest.” The time unit for Extra Tree is missed.

10. “Therefore although the overall accuracy of Extra Trees is higher 214. than the remaining algorithms, the training and testing time is very slow indicating a higher amount of resource consumption hence in general, Bagging could be the best suitable model both in terms of accuracy and speed.” Here the authors may need to discuss why a two second gap is so important?

11. Why these four models are used? Why not other forms? How do these compare to other machine learning models

12. Add discussion section.

13. The authors should provide a academic contribution discussion in the conclusion part.

14. Some of the references are outdated.

Reviewer 1 ·

Basic reporting

I have gone through the manuscript A remote sensing and ML based thoroughly investigated time series analysis of the global marine phytoplankton dynamics (#90310).

The author has done very good work, but this manuscript still needs more attention from the author.

Small errors in the language of the manuscript need to be revised.
Introduction section: The introduction needs to be supported by adding a literature review about using ocean color and advanced remote sensing data to monitor phytoplankton dynamics. Please add some specific literature. I think the authors could read them and cite them. please see

Abou Samra RM, El-Gammal M, Eissa R (2021) Oceanographic factors of oil pollution dispersion offshore the Nile Delta (Egypt) using GIS. Environmental Science and Pollution Research, 28:25830-25843 https://doi.org/10.1007/s11356-021-12570-0
Abou Samra R.M. & Ali R. (2022). Monitoring of oil spill in the offshore zone of the Nile Delta using Sentinel data. Marine Pollution Bulletin, 179, 113718.
The resolution of the figures should be enhanced, and the parameters should be written correctly.
Line 149, page 2 What does this abbreviation, CPU, refer to?

Experimental design

The methodology section should be well written. For example, what is the software used for processing the data?

Validity of the findings

Please compare your results with previous studies.

Additional comments

No additional comments

Reviewer 2 ·

Basic reporting

The article is self-contained, the authors performed several experiments to assess the use of ML algorithms to predict chlorophyll concentrations in different regions of the ocean. Relevant results are summarized in tables and figures. However, captions in figures are not self-explanatory and explanations in the main text are often deficient. There is also room for improvement in the language and sentence construction. While it is often easy to guess what the authors meant, the language is sometimes unclear or confusing. There are typos, particularly with punctuation marks out of place. Concepts of biological oceanography and phytoplankton are shallow and not well explained, and the authors appear unaware of the growing amount of literature concerning the use of ML in ocean biogeochemical models.

Experimental design

The study presented is original and within the aims and scope of the journal. The research question is straightforward; however, it is not well justified in context of existing literature about ocean biogeochemical models. The scope of the study is sometimes ambiguous. For example, the manuscript states in the abstract that “The state-of-the-art works have not shown much progress in utilizing remote sensing reanalysis data of oceanographic parameters for monitoring vertical variability of global marine phytoplankton.”, which seems to suggest this study will consider variability in the vertical structure of phytoplankton. However, later it is mentioned that the depth level analyzed is 0.5m, if I understood correctly.

The introduction appears to suggest that ML algorithms could replace satellite ocean color algorithms. However, this may be possible, the data set used to train the model are results from a dynamical ocean biogeochemical model in itself, thus results are not truly independent, and the experiments do not test the use of ML algorithms to estimate ocean color or satellite chlorophyll. The fact that he training data are model results themselves, partly explains the high correlation that the predictive ML algorithms are able to achieve. This may not be the same using in situ observations, which are necessary to build satellite ocean color algorithms. Rather that referring to the ML algorithm being an alternative to ocean color, the study should emphasize the use of ML approaches to reduce computational cost in dynamical models by offering alternative nested parameterizations instead of explicit prognostic ecosystem dynamics in global models. I believe there is growing literature with respect to this topic. It is also necessary to discuss biases and sources of error in both the training dataset and the ML results. Particularly, dynamical models tend to drift over time, thus a ML algorithm overfitted to a certain period may not be useful for forecasting.

I think the experiments are straightforward but are not well explained (probably due to a language limitation). Somewhere in the text it sounds like the Indian Ocean was the only data not used in the training data set, which would correspond to 20% of the total data. Later, experiments where each region was excluded from the training dataset are described. I also suggest including the boundaries of regions in the map (Fig 1), and being more clear in the captions of this figure (and all other figures in general).

I am also missing a discussion about the importance of each variable in the resulting ML algorithm. The ML algorithm, in this sense, is basically a black box. For example, the normalized nitrate values appear very low, which I guess is due to the normalization. Surface nitrate has a wide range of values, with both high geographical and temporal variability, thus the importance of nitrate may be underestimated in these ML algorithms. Nitrate is a very important nutrient and is usually the nutrient limiting phytoplankton growth in most regions. An increase in nitrate concentrations, brought to the surface during deep mixing events/periods is one of the important drivers of the seasonal cycle of phytoplankton. This seasonal cycle, which this study describes in the first paragraph of the results, is a well-known feature of temperate oceans. Authors should get better informed about phytoplankton phenology and spring blooms, to better support their discussion. For instance, how do the results from ML algorithms than the seasonal cycle? Predicting the seasonal cycle is easier than intra-seasonal variability, which poses a real challenge as phytoplankton concentrations do not respond linearly to nutrient, temperature, currents, etc (see for instance Barton et al., 2015 Physical controls of variability in North Atlantic phytoplankton communities).

Validity of the findings

The impact and novelty of this study should be better assessed and justified in the context of existing literature about phytoplankton phenology, biogeochemical modeling and ML in ocean models. The data used in the study is robust and cited, however it should be better understood what this dataset represents (model results for the biogeochemical variables and data-assimilative model results for the physical variables). These improvements will help grounding the study conclusions.

---

## Round 0.2 · Major Revisions

Your manuscript has been re-assigned to me as editor. Thank you for thoroughly revising the paper. However, a many issues remain on the writing. I'll point them down below:

1) Title and abstract still have abbreviations that were not previously defined. Notably, ML and AI.

2) Some sentences throughout the seem just odd. Like L. 47, which has too many variations of "detected".

3) Avoid making citations all at the end of paragraphs, such as the 1st paragraph of introduction.

4) remove the subheading 2 and integrate its content in the introduction. But try to summarize it to what is needed to understand the knowledge gap you're trying to fulfill. Most of its content looks like a literature review that you repeatedly list what specific papers have done and found. This is bad practice and must be avoided.

5) change 3. research methods to just Material and Methods

6) remove subheadings from the Discussion

Below you'll find a line by line list of comments. Do follow them closely and make all necessary adjustments.

L. 54: citation lacking
L. 59 "a lot" is informal language. Also, citation lacking
L. 62-3: something is lacking in this sentence. Seems unfinished
L. 62-9: these sentences are short and seem too telgraphic. Rephrase them to make the text more fluid
L. 69-70: the transition from one paragraph to the next is too abrupt. Again, improve text flow
L. 77 with -> which
L. 81-2: delete
L. 91: I develop -> i) develop
L. 96-100: these sentences are incomprehensible
L. 101: what do you mean by "to measure the dynamics of these features"?
L. 101-2: rephrase to "to our knowledge, no study has been forecasted marine phytoplankton distribution at a global scale"
L. 103-10: delete
L. 112: delete.
L. 139 & 155 & 171. you repeatedly uses the expression "In the state of the art" in a way that doesn't make sense at all. Revise it
L. 175: delete
L. 188-196: Delete. no need to provide such a detailed description of the dataset. Just refer the reader to the reference that does it.
L. 208-10: delete the geog coordinates and cimply cite Fig. 1
L. 210-1: delete
L. 214-7: delete, along with Fig. 2.
Table 1 is purely descriptive and must go to the suppl mat.
L. 219-37: provide many more citations here to back this statements up

-->> L. 241-2: This is important. This might also explain why the data in Fig. 7 and 8 look so bizare with many outliers and this . This 'normalization' procedure doesn't make sense at first sight. What you have in your models are predictors that were measured in different scales. Therefore, the best thing to do it standarize them to have 0 mean and unit sd, but not to vary between 0 and 1

L. 253-6: delete

-> L. 257-9: This is crucial: this looks more like a fishing expedition to me and it's in contrast with L. 94-5 where you mention you only used those four algorithms.

-->> L. 263-72: delete. No need to fully describe how each and every method works. This is not the central part of the paper. Simply refer the reader to a good ref . Instead, provide more details on how you parameterized each model and more importantly: how you evaluated performance of different models? R2 is overly emphasized in the Results, but it's not a good statistics to evaluate model performance. **Calculate AUC or ROC curve instead and rank models according to this criterion.**

L. 275-80: delete and see comment above.
L. 282-3: delete
L. 287-9: delete

-->> This is important: how you dealt with temporal and spatial autocorrelation in the models, which inflates degrees of freedom?

L. 312: you mean R^2 = 0.963142?
L. 313-4: you already said it in the Methods that this applies to all models.

Fig. 4 and Fig 5: this shows the result of which algorithm? Also, what do you mena by "parts of the ocean"?

L. 315: "pretty impressive" is too informal.

L. 315-6 & 322-3: these sentences don't make any sense whatsoever

L. 324-6: these very high R2 are suspicious and make me think of overfitting

L. 341-5. move this to the legend of figure 6.

L. 361-5: delete. Cite the tables appropriately in the following paragraph

L. 394-5: delete

L. 397: this sentence doesn't make any sense "The study surpasses the state of the art in multiple areas." , as does the title of this subheading

L. 398: you can't refer to regions in the ocean as "water bodies". Correct this throughout the text.

L. 399-408: again, the Discussion section requires you to establish a dialogue with the literature, to contrast your results with previous studies, and not simply lists what other studies found. This is no Discussion.

Fig. 7 and 8. The legends of these figures don't make sense, as much as the data as much as the abbreviations in the X axis, which are not explained.

L. 410-5: delete

L. 424: delete

L. 429-33: these sentences don't make any sense.

L. 434-50: delete

L. 453-7: delete

In your response letter, please, be straightforward and as much direct as possible. Avoid saying "thank you" and use elaborated wordings in each and every response.

Reviewer 1 ·

Basic reporting

The revised manuscript is acceptable in its current form from the scientific point of view.

Experimental design

Well written and acceptable.

Validity of the findings

Well written and acceptable.

---

## Round 0.3 · Major Revisions

I'm sorry, but I cannot fully evaluate your paper without detailed responses to each of my previous comments. Responses like this "The sentences have been reframed and rewritten to make them meaningful." or even "The said sentences have been revised with more appropriate terms." force me to go again through all the paper again to actually check and see if you have made all the changes.
Please, refer to the template available at PeerJ's website and instructions on how to properly respond to a decision letter.

---

## Round 0.4 · accepted · Accept

Thank you for carefully revising the manuscript following my suggestions. I'm glad to recommend it for publication as is.